# Sodium Butyrate Supplementation in Whole Milk Modulates the Gastrointestinal Microbiota Without Altering the Resistome and Virulome in Preweaned Calves

**DOI:** 10.3390/microorganisms13102375

**Published:** 2025-10-15

**Authors:** Liwen Xing, Song Niu, Donglin Wu, Zhanghe Zhang, Ming Xu

**Affiliations:** College of Animal Science, Inner Mongolia Agricultural University, Hohhot 010018, China; 13604718981@163.com (L.X.); ndns@emails.imau.edu.cn (S.N.); wdl2017@emails.imau.edu.cn (D.W.)

**Keywords:** calf feeding, sodium butyrate, metagenomic sequencing, rumen microbiota, cecum microbiota

## Abstract

This study investigated the effects of supplementing whole milk (WM) with sodium butyrate (SB) on the gastrointestinal microbiota of preweaned calves. Twelve newborn Holstein calves (4 days old, 39.21 ± 1.45 kg) were randomly assigned into one of two dietary treatments: (1) WM without SB (CON) and (2) WM supplemented with SB (8.8 g/d; SB). At 74 days of age, all calves were slaughtered to collect ruminal and cecal digesta. Metagenomic analysis was used to characterize the microbial composition, virulence factor genes (VFGs), and antibiotic resistance genes (ARGs). SB supplementation altered the ruminal microbial composition and increased the abundance of beneficial bacteria, including *Actinobacteria*, *Bifidobacterium*, and *Olsenella* (*p* < 0.05). Although SB did not significantly affect the overall microbial composition or diversity in the cecum (*p* > 0.05), it promoted the growth of beneficial genera such as *Flavonifractor* and *Subdoligranulum* (*p* < 0.05). Furthermore, SB supplementation did not significantly alter the composition of VFGs or ARGs in either the rumen or the cecum (*p* > 0.05). However, significant differences in VFGs and ARGs were observed between the rumen and the cecum, with a greater diversity of both VFGs and ARGs detected in the cecum compared to that in the rumen (*p* < 0.05). In summary, supplementing whole milk with sodium butyrate modulates gastrointestinal health in preweaned calves by favorably shaping the microbial community without significantly altering the antibiotic resistome or virulome.

## 1. Introduction

Antimicrobial drugs are commonly used in livestock production for both disease prevention and growth enhancement. Within this context, dairy cows are of particular importance due to their specialized gastrointestinal system, wherein the rumen harbors a complex microbial ecosystem that serves as a reservoir for virulence factor genes (VFGs) and antibiotic resistance genes (ARGs) [1,2]. ARGs can not only persist internally but are also excreted via manure, substantially contributing to environmental contamination and creating potential pathways for dissemination into soil, water, and the food supply [3,4]. It has been demonstrated that the gastrointestinal tract of newborn ruminants is initially sterile [5] but becomes rapidly colonized by microorganisms derived from the dam’s vagina, colostrum, and the external environment [6]. As microbial composition and diversity in the gut are closely linked to the profiles of VFGs and ARGs [7,8], the critical influence of gastrointestinal microbiota on host health, coupled with the environmental risks posed by ARGs, necessitates the exploration of effective strategies for their modulation.

Among potential modulators, butyric acid—a short-chain fatty acid (SCFA) that participates in key host metabolic pathways—has attracted significant interest. This compound has been investigated for its ability to ameliorate intestinal pathologies, including neoplasia, inflammatory bowel disease, and malabsorptive states, underscoring its potential for clinical application [9]. Due to the volatile and unstable nature of butyric acid, it is frequently administered in feed as the more stable sodium butyrate (SB). Investigations into SB in animal science date back to the late 1980s, culminating in its inclusion in China’s “Catalog of Feed Additives” in 2003 [10]. Subsequent research has established its positive effects in diverse species, where SB enhances growth performance [11,12], supports the development of the GIT [11,13,14] and the pancreas [15], and regulates microbial colonization [10,16]. The mechanistic basis for these benefits was partly elucidated by a study showing that SB ameliorates induced colitis and epithelial barrier dysfunction via G Protein-Coupled Receptor 109A activation and inhibition of the AKT and TNF-α p56 pathways [17]. Thus, evidence suggests SB can reduce inflammation and promote systemic health. Our earlier work in calves established that supplementing milk with SB at 8.8 g/d promoted beneficial gastrointestinal microbial development [18,19], an effect corroborated by studies in other animal models [20,21,22].

Given this background, we hypothesized that long-term SB supplementation in whole milk would not only modulate the gastrointestinal microbiota of preweaned calves but also exert a concomitant impact on the resistome and the virulome by reducing the abundance of VFGs and ARGs. The present study was therefore designed to directly assess the effect of SB on these microbial and genetic profiles, aiming to provide insights into its potential to optimize gut health and mitigate antimicrobial resistance risks.

## 2. Materials and Methods

### 2.1. Animals, Treatments, and Diets

A total of 12 newborn Holstein calves born between 6 July and 9 August 2023 from multiparous cows were used in this study. The trial was conducted from 6 July to 22 October 2023. Calves were housed individually in hutches (2.6 × 1.5 m) bedded with dry sand. At 24 h post-birth, blood was collected from the external jugular vein to determine serum immunoglobulin G (IgG) concentration, ensuring successful passive immunity transfer. The average serum IgG concentration was 23.64 ± 4.32 mg/mL, with no significant differences between treatment groups. At 4 days of age, the calves were randomly assigned into one of two treatments in a completely randomized design: (1) the control (CON), receiving whole milk without supplementation, or (2) sodium butyrate (SB), receiving whole milk supplemented with 8.8 g/d of SB. The SB product (Jiabaoyu; ≥98% purity) was supplied as a raw powder by Jinan Degao Agriculture and Animal Husbandry Technology Co., Ltd. (Jinan, China).

Immediately after birth, each calf received 4 L of pasteurized colostrum, followed by an additional 2 L after 6 h. Thereafter, calves were fed pasteurized whole milk (WM) according to the following schedule: 4.4 L/d from days 4 to 14, 8 L/d from days 15 to 28, and 8.8 L/d from days 29 to 64. The milk, sourced from the farm, was pasteurized prior to feeding. Calves were fed twice daily at 05:00 and 17:00 using individual buckets, which were thoroughly cleaned and dried after each use to maintain hygiene. A commercial pelleted starter (diameter: 3.1 mm; Tianjin Jiuzhou Dadi Feed Co., Ltd., Tianjin, China) was introduced on day 3 to stimulate rumen development. To facilitate a smooth dietary transition, a 10-day weaning period was implemented, with complete weaning occurring at 74 days of age. Samples of the pelleted feed and WM were collected weekly, pooled, and stored at −20 °C for subsequent nutritional analysis (Appendix A), which was performed according to established methods [11,23]. No antibiotics were administered throughout the trial.

### 2.2. Sample Collection from the GIT

At 74 days of age, following a 4 h fast, the twelve male calves were euthanized through exsanguination. Immediately after slaughter, the rumen and the cecum were isolated and opened. The digesta from each compartment was collected separately, thoroughly mixed, and promptly aliquoted into 2 mL cryotubes. All samples were then flash-frozen in liquid nitrogen for subsequent metagenomic analysis.

### 2.3. DNA Extraction, PCR Amplification, and 16S rRNA Sequencing

Total genomic DNA was extracted from rumen and cecum samples using the E.Z.N.A.^®^ Soil DNA Kit (Omega Bio-tek, Norcross, GE, USA) according to the manufacturer’s instructions. The DNA extract was checked on 1% agarose gel, and DNA concentration and purity were determined with a NanoDrop 2000 UV-vis spectrophotometer (Thermo Scientific, Wilmington, DE, USA). The hypervariable region V3–V4 of the bacterial 16S rRNA gene was amplified and then sequenced with the Illumina MiSeq platform (Majorbio BioPham Technology, Shanghai, China) using the primers 338F (5′-ACTCCTACGGGAGGCAGCA-3′) and 806R (5′-GGACTACHVGGGTWTCTAAT-3′). PCR was performed in triplicate under the following conditions: initial denaturation at 95 °C for 3 min and then 29 cycles of 95 °C for 30 s, 55 °C for 30 s, and 72 °C for 45 s, followed by a final extension at 72 °C for 10 min. The PCR mixture included 2 μL of 5 × PrimerSTAR buffer (4 μL), dNTPs (2.5 mM), 2 μL of the forward primer (5 μM), 0.8 μL of the reverse primer (5 μM), 0.8 μL of PrimerSTAR heat stress DNA polymerase, and 20 ng of template DNA. Two percent agarose gels were used to detect the PCR products, which were then purified using a DNA purification kit (Axygen, Biosciences, Union City, CA, USA). The raw 16S rRNA sequences were demultiplexed, quality-filtered using fastp version 0.20.0, and merged using FLASH version 1.2.7, and the quality sequences were filtered using previously reported criteria [24]. Operational taxonomic units (OTUs) with a 97% similarity cutoff were clustered using UPARSE (version 7.0; http://drive5.com/uparse/, accessed on 20 July 2025), and chimeric sequences were identified and removed [25]. The taxonomy of each OUT was assigned by classifying its representative sequence using the PDR Classifier algorithm (http://www.arb-silva.de/, accessed on 29 July 2025) using a confidence threshold of 70% [26]. To contextualize our findings, the linear discriminant analysis effect size (LEfSe) [27] was calculated to investigate the differential microbial abundance with the SB supplementation methods.

### 2.4. Metagenomics Sequencing

DNA extracts were fragmented to an average size of about 400 bp using Covaris M220 (Gene Company Limited, Hong Kong, China) for paired-end library construction. The paired-end library was constructed using NEXTflexTM Rapid DNA-Seq (Bioo Scientific, Austin, TX, USA). Adapters containing the full complement of sequencing primer hybridization sites were ligated to the blunt end of the fragments. Paired-end sequencing was performed on the Illumina NovaSeq/Hiseq Xten (Illumina Inc., San Diego, CA, USA) at Majorbio Bio-Pharm Technology Co., Ltd. (Shanghai, China) using NovaSeq Reagent Kits/HiSeq X Reagent Kits (Illumina Inc., San Diego, CA, USA) according to the manufacturer’s instructions (www.illumina.com, accessed on 15 May 2025). The raw reads from metagenome sequencing were used to generate clean reads through removing adaptor sequences, trimming, and removing low-quality reads (reads with N bases, a minimum length threshold of 50 bp, and a minimum quality threshold of 20) using fastp [28] (https://github.com/OpenGene/fastp, accessed on 15 May 2025, version 0.20.0) on the free online platform Majorbio Cloud Platform (cloud.majorbio.com, accessed on 22 July 2025). These high-quality reads were then assembled into contigs using MEGAHIT [29] (parameters: kmer min = 47; kmer max = 97; step = 10) (https://github.com/voutcn/megahit, accessed on 22 July 2025, version 1.1.2), which makes use of succinct de Bruijn graphs. Contigs with a length of 300 bp or over were selected as the final assembling result. To assess the effects of NAR supplementation on antibiotic resistance and virulence factors, microbial functions were analyzed using the Comprehensive Antibiotic Resistance Database (CARD, v3.0.9; https://card.mcmaster.ca/, accessed on 20 July 2025) and the Virulence Factor Database (VFDB; https://www.mgc.ac.cn/VFs/main.htm/, accessed on 20 July 2025). Contigs were aligned with CARD using DIAMOND with an E-value threshold of ≤1 × 10^−5^, only retaining hits with a query coverage > 90% and a sequence identity > 70% for ARG annotations. Similarly, VFs were annotated against VFDB using DIAMOND with the same *E*-value threshold. The relative abundances of ARGs and VFGs were calculated using TPM (Transcripts Per Million), which normalized for sequencing depth and gene length, enabling comparisons between samples. To identify potential hosts of ARG- and VF-associated contigs, these sequences were further taxonomically classified against the nucleotide (NR) database.

### 2.5. Statistical Analysis

Based on Górka et al. [30], PROC POWER and the GLMPOWER procedure in SAS (version 9.2; SAS Institute Inc., Cary, NC, USA) were used to yield a sample size with a power of 0.8 under *p* < 0.05. According to previous similar studies, the effect sizes f that made a difference in BW, diarrhea frequency, and crypt depth were 0.68 [31], 1.30 [32], and 1.24 [33], respectively. Based on the results of the power analysis, a total of at least 20 calves (10 calves per treatment) were needed to study growth performance, and 8 calves (4 calves per treatment) were needed to study health and intestinal development to meet the power requirements with treatment groups of 2. The 12 male calves that were slaughtered were used to analyze the data on the gastrointestinal tract microbiota on d 74. Therefore, selecting 6 replicates per group in this study was deemed reasonable.

Alpha diversity was assessed using various indices, such as community diversity (Shannon index, Simpson index) and community richness (Chao1 index). For beta diversity (diversity between samples), the weighted UniFrac distance was used, followed by Adonis analysis, and visualization was performed using non-metric multidimensional scaling (NMDS). Adonis is a nonparametric technique that tests differences in community structures among populations. Unique and core microorganisms of the genus were depicted using Venn diagrams. Dominant bacteria at both the phylum and genus classifications were identified as those representing a prevalence of ≥0.1% in all groups examined. These core bacterial populations were subsequently subjected to rigorous statistical analysis. Bacterial linear discriminant analysis effect size (LEfSe) was calculated using the online platform Majorbio Cloud Platform (www.Majorbio.com). The LEfSe analysis was used to compare significant differences in bacterial composition between the different treatments. A Kruskal–Wallis test was performed to calculate the significance *p* value in the LEfSe analysis, followed by multiple comparisons adjusted by the false discovery rate (fdr). Before statistical analysis, data were checked for homogeneity of variance and transformed as needed. Microbiota-related indices (alpha diversity index, taxonomic analysis at the phylum and genus levels) were analyzed through one-way ANOVA using SPSS statistical software (version 24.0; SPSS Inc., Chicago, IL, USA). We then refined the mean differences using the Tukey–Kramer post hoc test. A *p* < 0.05 value indicated a significant difference.

## 3. Results

### 3.1. The 16S rRNA Sequencing Analysis and Quality Evaluation

Differences in the microbial compositions of the rumen and the cecum of lactating calves were evaluated using 16S rRNA gene sequencing techniques. Overall, 1,260,945 clean reads were obtained from 24 samples (12 rumen and 12 cecum fluid samples); the Good’s coverage indices in all samples were greater than 99%, indicating that the majority of the bacteria present in the samples were identified and that the sequencing depth was adequate for community analysis. Thus, the data were sufficient to analyze the microbial communities. Additionally, 522,876,627 bases and 415 average sequence lengths were detected (Appendix A).

### 3.2. Diversity and Composition Analysis

In the rumen, bacterial β-diversity indices exhibited some variation. Non-metric multidimensional scaling (NMDS) based on clustering at the genus level showed that the bacterial community composition in the samples was clearly separated between the CON and SB groups (*p* < 0.05, Figure 1A). Venn’s analysis showed that there were 136 (4.57%) genera specific to the SB group, suggesting that the addition of SB changed the composition of the rumen at the genus level (Figure 1B). The species abundance at various taxonomic levels, including phylum and genus, was assessed and ordered. At the phylum level, the rumen was predominantly populated by *Firmicutes* and *Bacteroidetes* (Figure 1C). At the genus level, the five most abundant genera were *unclassified_f__Lachnospiraceae*, *Prevotella*, *Olsenella*, *unclassified_o__Clostridiales*, and *Intestinibaculum* (Figure 1D). Student’s *t*-test results indicated that the addition of SB increased the abundance of *g_Bifidobacterium*, *g_Fusobacterium*, and *g_Lactimicrobium* and decreased the abundance of *g_Bacteroides* and *g_Phocaeicola* in the rumen as compared to that in the CON group (*p* < 0.05; Figure 1E). Furthermore, using the linear discriminant analysis (LDA) (Figure 1F), three bacterial phyla, four bacterial classes, three bacterial orders, six bacterial families, and four bacterial genera were identified in both groups, and the addition of SB to WM resulted in a significant enrichment in *g_Bifidobacterium* and *g_Olsenella* in the SB group.

No effect of the addition of SB to WM on cecum bacterial α- and β-diversity was found in this study (*p* > 0.05, Appendix A). Venn’s analysis showed that there were 234 (7.07%) genera specific to the SB group, suggesting that the addition of SB changed the composition of the rumen at the genus level (Figure 2B). The species abundance at various taxonomic levels, including phylum and genus, was assessed and ordered. At the phylum level, the rumen was predominantly populated by *Firmicutes* and *Bacteroidetes* (Figure 2C). At the genus level, the five most abundant genera were *unclassified_o__Clostridiales*, *Bacteroides*, *Prevotella*, *unclassified_p__Firmicutes*, and *unclassified_f__Lachnospiraceae* (Figure 2D). Student’s *t*-test results indicated that the addition of SB increased the abundance of *g_Flavonifractor*, *g_Subdoligranulum*, *g_Colidextribacter,* and *g_Agathobaculum* and decreased the abundance of *g_Agathobacter* in the cecum as compared to that in the CON group (*p* < 0.05; Figure 2E). Furthermore, using LDA (Figure 2F), two bacterial phyla, two bacterial classes, two bacterial orders, one bacterial family, and three bacterial genera were identified in the groups, and the addition of SB to WM resulted in significant enrichment in *g__unclassified_f__Rikenellaceae* and *g__Subdoligranulum* in the SB group.

### 3.3. The Distribution of VFGs and ARGs Along the GIT of Preweaned Calves

Additionally, we conducted an in-depth analysis to examine the impact of adding SB to WM on VFGs in samples taken from the rumen and the cecum of preweaned calves. We initially noticed that there were significant differences in the overall composition of VFGs between the rumen and cecum samples (*p* < 0.05; Figure 3A). However, our main focus was on how SB supplementation influenced the VFGs in these two gastrointestinal regions.

In the rumen, we found that SB treatment did not alter the composition and diversity of VFGs. Based on classification at functional level 1, offensive VFGs were the most prevalent among the four groups, followed by defensive VFGs, nonspecific VFGs, and those involved in the regulation of virulence-associated genes (Appendix A). At level 2, the top eight VFG functions were adherence, antiphagocytosis, iron uptake, regulation, toxin production, secretion, serum resistance, and stress response proteins (Appendix A). Using LEfSe (LDA > 3; *p* < 0.05) to identify signature VFGs in each group, we discovered that in the control (CON) rumen, Pse5Ac7Ac, Pse5Ac7Am, Pse8OAc, Pse5Am7AcGlnAc (AI151), SenX3 (CVF666), LOS (VF0044), Type IV pili (VF0082), PhoP (VF0286), type IV pili (AI117), Mip (VF0153), and CdpA (VF0432) were the characteristic VFGs. In contrast, for the SB-supplemented rumen, the VFGs Capsule (CVF186), FbpABC (VF0272), lateral flagella (AI142), repeat in toxin (RTX) (CVF795), Oligopeptide-binding protein (CVF240), HitABC (VF0268), Hydrogen cyanide production (CVF549), Exopolysaccharide (CVF495), Heme biosynthesis (CVF506), and Pyochelin (VF0095) were more abundant. This indicates that although SB did not change the overall composition and diversity, it did lead to a shift in the specific VFGs present in the rumen (Figure 3C).

Similarly, in the cecum, SB treatment also had no effect on the composition and diversity of VFGs. At functional level 1, the distribution pattern of offensive, defensive, and nonspecific VFGs, as well as those involved in the regulation of virulence-associated genes, was consistent with that in the rumen (Appendix A). The top eight VFG functions at level 2 were the same those as in the rumen: adherence, antiphagocytosis, iron uptake, regulation, toxin production, secretion, serum resistance, and stress response proteins (Appendix A). Through the LEfSe analysis, we identified distinct signature VFGs for the CON and SB cecum groups. In the CON cecum, the signature included AdeFGH efflux pump (VF0504), Fibronectin-binding protein (AI238), Capsule (VF0560), Alginate (VF0091), MtrCDE (VF0451), LPS (VF0033), Cytolysin (VF0356), and polar flagella (AI149). In the SB cecum group, Pyrimidine biosynthesis (VF0558), RelA (VF0287), Capsule (VF0003), ClpC (VF0072), Acinetobactin (VF0467), mu-toxin (VF0389), RegX3 (CVF667), Alginate regulation (CVF523), PDIM (phthiocerol dimycocerosate), PGL (phenolic glycolipid) biosynthesis and transport (CVF288), and GacS/GacA two-component system (CVF529) comprised the signature. This shows that SB supplementation, despite not affecting the overall composition and diversity, also produced a change in the specific VFGs in the cecum (Figure 4C).

We conducted a comprehensive analysis to elucidate the effect of SB supplementation on the ARGs in the rumen and the cecum of preweaned calves. It was observed that there were significant differences in the composition of ARGs between the rumen and cecum samples (*p* < 0.05; Figure 4A), with the cecum exhibiting a higher overall abundance of ARGs compared to that in the rumen (*p* < 0.05; Figure 3B). However, our primary focus was on the specific responses of each gastrointestinal compartment to SB treatment. When ARGs were categorized according to their resistance mechanisms, a consistent pattern emerged across both the rumen and the cecum. Antibiotic efflux ARGs were the most dominant among the four identified resistance mechanisms in all groups. This was followed by antibiotic target alteration ARGs, antibiotic target protection ARGs, antibiotic inactivation ARGs, and antibiotic target replacement ARGs (Appendix A). This suggests that the fundamental mechanisms of antibiotic resistance in these gastrointestinal regions are broadly similar in preweaned calves, regardless of SB treatment. In terms of antibiotic class, the top eight ARG functions were also consistent between the rumen and the cecum. These classes included multidrug, MLS, glycopeptide, tetracycline, peptide, aminoglycoside, aminocoumarin, and fluoroquinolone (Appendix A). This indicates a shared set of antibiotic-resistance-related functions in these two parts of the calf’s GIT. Interestingly, SB treatment did not cause any significant changes in the overall composition and diversity of ARGs in the rumen. This implies that the general structure of the ARG community at this site remained relatively stable under SB supplementation. However, when we employed LEfSe analysis (LDA > 3; *p* < 0.05) to identify signature ARGs in each group (Figure 4C), distinct differences were observed. In the control (CON) rumen, signature ARGs such as vanSM, arlR, tetQ, arlS, MuxA, tetS, mecB, vanHA, Bifidobacterium adolescentis rpoB mutants conferring resistance to rifampicin, and farA were identified. These genes likely represent the characteristic ARGs in the rumen of preweaned calves under normal conditions. In contrast, the SB-supplemented rumen showed a different set of abundant ARGs. Genes such as macB, efrA, patB, patA, oleC, tetW, lmrD, tetB(60), Staphylococcus mupA conferring resistance to mupirocin, and vanHB were more prevalent. This shift in ARG signature suggests that SB treatment may influence ARGs that are highly expressed or enriched in the rumen, even though the overall composition and diversity remain unchanged. Similarly to observations in the rumen, SB treatment had no effect on the composition and diversity of ARGs in the cecum. The cecum’s ARG community structure appeared to be resilient to SB supplementation in terms of these broad-scale characteristics. The LEfSe analysis revealed distinct signature ARGs in the cecum in the CON and SB groups. In the CON cecum, APH(3″)-Ib, cmeB, ANT(6)-Ib, adeK, rpoB2, adeS, ErmF, ceoB, vanSE, and vmlR were enriched, constituting the signature ARGs. These genes likely play important roles in the antibiotic resistance profile of the cecum under normal conditions. In the SB-supplemented cecum group, a different set of signature ARGs emerged, including ErmB, CfxA6, mefC, kdpE, vanRI, vanG, APH(3′)-IIIa, ugd, LpeB, and Agrobacterium fabrum chloramphenicol acetyltransferase. This indicates that SB treatment, despite not affecting the overall composition and diversity, is associated with changes in ARG expression within the cecum.

## 4. Discussion

There is increasing evidence that SB can have beneficial effects on gastrointestinal microbes in pigs [16,34]. More than a decade ago, Li et al. [35] reported that SB supplementation increased butyrate production by increasing the abundance of *Firmicutes* and decreasing that of *Bacteroidetes* in dairy cows. Our recent publication demonstrated that the optimal supplementation of SB in milk prior to weaning (45 days of age) was 8.78 g/d [36]. Based on this, the present study extended the supplementation of SB to the entire suckling period.

The α-diversity index is used to directly illustrate microbiota abundance and richness. In the present study, it was found that SB supplementation had no effect on the α-diversity index in both the rumen and the cecum. Similarly, another study found that SB supplementation did not affect rumen α-diversity [37]. In contrast, a further study found that SB supplementation did affect α-diversity in the rumen [18]. This may be because the rumen is not fully developed in the early stages of life. In the present study, it was found that SB supplementation significantly affected rumen β-diversity (NMDS), indicating that SB supplementation significantly altered the composition of microorganisms at the genus level at this site. Similar results were observed in a recent study, where the group fed with SB-supplemented milk exhibited significantly increased rumen microbial diversity compared to that in the CON group [18]. This is an interesting finding because milk bypasses the rumen due to the reflexive closure of the reticular (or esophageal) groove when calves nurse. Similar results were reported regarding rumen microbial diversity in calves receiving milk supplemented with SB in previous studies [18,37]. However, SB supplementation was shown to have no effect on cecum α- and β-diversity in the present study. This may be because as calves become older, exogenous supplementation of SB has limited influence on their gut microbiota. Related studies indicate that the optimal SB supplementation level exhibits a linear decreasing trend with calf age [18,36], suggesting that SB supplementation during the late suckling period may be redundant.

In the present study, supplementing WM with SB was found to significantly increase the abundance of *Actinobacteriota* in the rumen. Moreover, a positive correlation was found between rumen *Actinobacteriota* abundance and yak calf body weight in another study [38]. In addition, in our previous study, we similarly found a linear relationship between *Actinobacteriota* and SB supplementation [18]. Numerous reports on SB supplementation have found that it can enhance growth performance in calves [11,18,30,36]. *Bifidobacterium* is closely linked to gut health and is commonly used as a health-promoting probiotic [39]. In this study, it was found that SB supplementation in WM significantly increased the abundance of *Bifidobacterium* in the rumen. In a recent study, it was shown that calves with a predominantly *Bifidobacterium* enterotype had a better growth performance and were healthier [40]. Furthermore, *Bifidobacterium* has the ability to directly convert starch into short-chain fatty acids (SCFAs) [40], which are critical in nurturing a balanced microbial habitat within the gut [41]. Additionally, these SCFAs are implicated in maintaining ruminal pH balance, stimulating the growth of the rumen wall, and modulating the host’s immune functions [42]. This indicates that *Bifidobacterium* plays a significant role in preweaned calves, potentially by maintaining rumen health and enhancing digestion. Members of the *Bacteroides* genus, which are prevalent anaerobic gut bacteria, are pivotal in carbohydrate and protein metabolism and play a vital role in the production of SCFAs [43,44], which are also integral for energy and modulating immune and inflammatory responses [45]. A significant decrease in the abundance of rumen *Bacteroides* after SB supplementation was found in the present study, whereas a quadratic effect with increasing SB supplementation was similarly found in our previous study [18]. This indicates that supplementing WM with short-chain butyric acid leads to a reduction in the abundance of bacteria producing anti-inflammatory short-chain fatty acids, thereby promoting calf health. The beneficial effect of SB on calves is further supported by the fact that SB supplementation was also found to reduce the inflammatory response in calves [11]. In our study, it was found that milk supplementation with SB increased the abundance of SCFA-producing bacteria, such as *Flavonifractor* and *Subdoligranulum,* in the cecum. The bacteria *Flavonifractor* and *Subdoligranulum*, which belong to the *Firmicutes* phylum, can produce butyric and acetic acid [46] and are negatively correlated with insulin resistance [47,48]. In one of our recent studies, it was found that supplementing milk with SB increased insulin secretion in calves [15]. SB supplementation has been found to increase insulin sensitivity in other studies [49]. This further demonstrates that SB supplementation can enhance calves’ digestive capacity, enabling them to achieve a superior growth performance.

Our study identified adherence, antiphagocytosis, and iron uptake as the main functions of VFGs in the rumen and cecum. Adhesion is crucial for pathogenic bacteria to infect the host; overcoming the repulsive force on the cell surface is the basis of successful adhesion. During this process, adhesion factors, including hemagglutinin, enolase, MAM-7, and T6SS2, interact with specific epithelial cell surface receptors and release toxins to infect host cells [50,51]. Previous studies have confirmed that exogenous mastitis in dairy cows is mostly caused by the adhesion of environmentally pathogenic microbes to breast tissue [52]. Bacteria’s antiphagocytic effect primarily occurs through their capsules, fimbriae, or secreted exotoxins, which can inhibit phagocytosis by host phagocytes and antibody-mediated immunity [53,54]. High-energy or low-pH diets can allow some low-abundance Clostridium in the GIT of healthy ruminants to proliferate rapidly and produce large amounts of exotoxin, inhibiting the immune response of host macrophages and altering gut permeability, thereby causing host dehydration and even death [55,56]. Interestingly, all calves in this study were healthy but had detectable VFGs in the GIT, suggesting that the calves’ GIT is a natural reservoir for VFGs, which is consistent with the findings of Zhuang et al. [7]. Although SB supplementation was not observed to affect rumen and cecum VFGs in this study, the composition and diversity of VFGs in the rumen and the cecum were significantly different. Cecum VFG diversity was higher than that in the rumen. This may be because the rumen is not fully developed in preweaned calves, when the intestine is an important site for nutrient digestion and absorption. In a previous study, it was found that the diversity of VFGs in the rumen of adult cows was significantly higher than that in feces, and, interestingly, the abundance and diversity of VFGs in feces were also significantly correlated with those in the rumen, suggesting that VFG expression is not isolated between different segments of the intestine and can be coordinated along the GIT [7].

Antimicrobial resistance (AMR) increases morbidity and mortality and is a severe global public health threat [57]. At the same time, the widespread use of antibiotics in the livestock husbandry industry has led to the enrichment and transfer of ARGs in the intestines and feces of livestock [58], which in turn leads to ARG contamination of the environment [59]. To date, resistome characterization has mostly been conducted on rumen contents and feces [60]. However, in the present study, SB supplementation was observed to have no effect on the rumen and cecum ARGs. However, there were significant differences in ARG composition between the rumen and the cecum, suggesting that ARGs are distributed differently in different spaces. Interestingly, significant differences in AMR abundance have been observed among animals of different ages in several species, including cattle [61,62,63,64]. Changes in the resistance genome structure and abundance of resistance genes in newborn calves up to seven weeks of age were demonstrated in a study proposing that colostrum is a source of resistant bacteria in the calf’s gut and that changes in the resistance genome are likely attributable to dietary changes [65]. The presence of heavy metals and minerals in forages, grains, mineral supplements, and milk replacer (particularly zinc, copper, chromium, arsenic, cadmium, and lead) may select for ARGs in the animal gut [66,67]. For example, zinc supplementation in diets has been suggested to enhance immune function, enhance feed efficiency, and aid in growth promotion [68,69,70], but the inadvertent selection of resistance genes may occur due to this practice, as has been observed in swine and poultry [71,72]. Metal resistance genes and ARGs are frequently collocated on plasmids, so the addition of metals to the gut can potentially select for organisms that encode both [73,74]. Notably, a high abundance was found in the GIT of young animals in the present study, suggesting that colostrum may be a major source of ARGs in the GIT of newborn calves. Further interventions are needed in the neonatal period, which may be a critical stage for reducing ARG expression. In this study, supplementation with SB did not affect the diversity or composition of rumen and cecal ARGs and VFGs, likely because the selected calves were in good health. Additionally, the supplementation of WM with SB directly acts on the gut, which may also contribute to its effects on rumen ARGs and VFGs. Notably, SB undergoes rapid degradation and absorption in the GIT, where it is oxidized into ketone bodies. Furthermore, the optimal SB supplementation rate decreases with calf age, potentially explaining why supplementation may not yield effects during the preweaning period.

In this study, we investigated the effects of SB supplementation on rumen and cecum microbiota, VFGs, and ARGs in preweaned calves. We found no significant impact of SB supplementation on virulence factors or resistance genes, but it did promote the colonization of beneficial microorganisms in the rumen and the cecum. These findings may be limited by the quality of colostrum received by calves and the small sample size. Future research plans will further evaluate nutritional strategies (such as supplementing with Bifidobacterium) to reduce the abundance of gastrointestinal pathogens and resistance genes in preweaning calves, providing theoretical support for healthy dairy cattle management. Additionally, this study found that supplementing milk with sodium butyrate affects rumen development, necessitating future investigation into the mechanisms underlying rumen–gut interactions.

## 5. Conclusions

In conclusion, supplementing whole milk with sodium butyrate enhanced the colonization of beneficial microorganisms in both the rumen and cecum of preweaned calves. While this supplementation significantly altered ruminal microbiota diversity, it did not affect the microbial diversity in the cecum. Furthermore, sodium butyrate had no significant impact on the abundance of antibiotic resistance genes or virulence factor genes in either the rumen or the cecum.

## Figures and Tables

**Figure 1 microorganisms-13-02375-f001:**
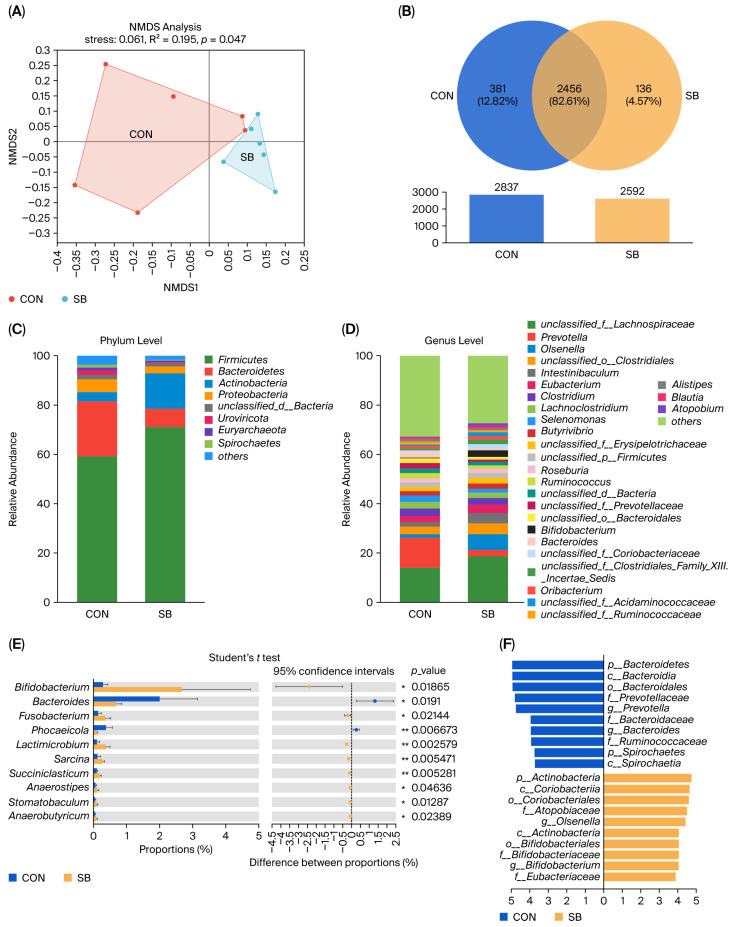
Effects of sodium butyrate supplementation in whole milk on rumen microbial communities in Holstein calves. (**A**) Non-metric multidimensional scaling (NMDS) plot of microbial communities at the genus level based on Bray–Curtis distance. (**B**) A genus-level Venn diagram illustrating microbial community overlap between groups. Rumen microbial composition was further analyzed at the phylum (**C**) and genus (**D**) levels. To identify differentially abundant species from the phylum to genus levels, two methods were employed: Student’s *t*-test (**E**) and a LEfSe bar chart (**F**). Differences were defined as significance with * *p* < 0.05, ** *p* < 0.01.

**Figure 2 microorganisms-13-02375-f002:**
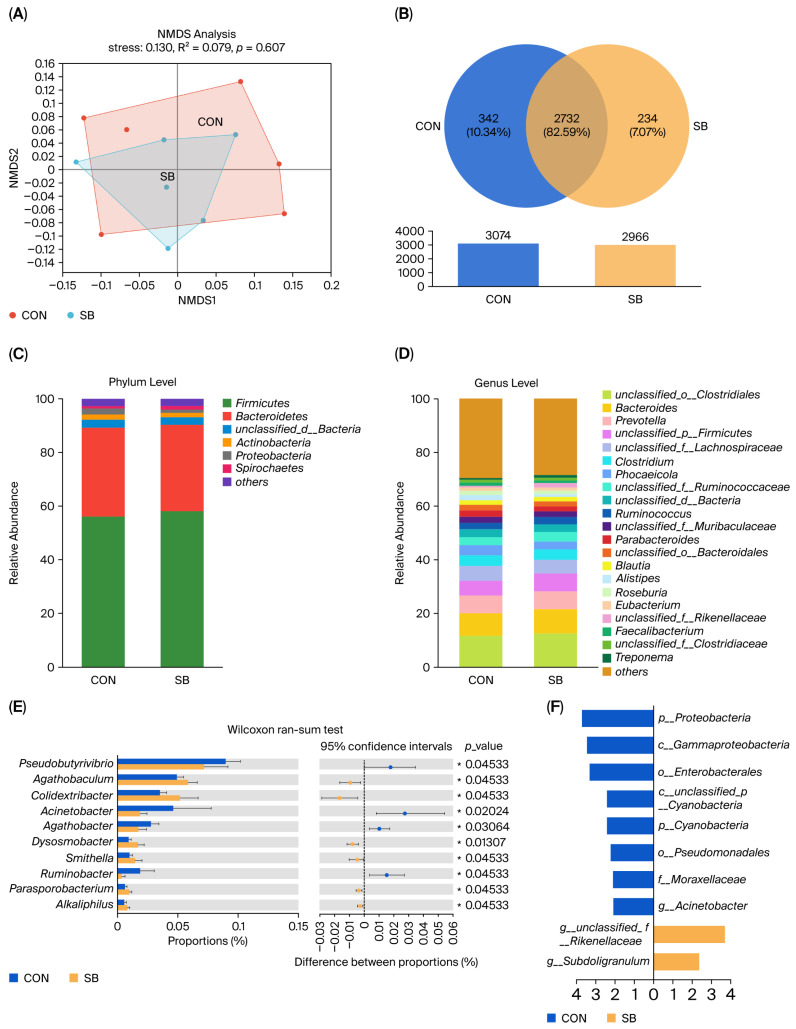
Effects of sodium butyrate (SB) supplementation in milk on cecum microbiota. (**A**) Non-metric multidimensional scaling (NMDS) plot of microbial communities at the genus level based on Bray–Curtis distance. (**B**) A genus-level Venn diagram illustrating microbial community overlap between groups. Cecal microbiota composition was further analyzed at the phylum (**C**) and genus (**D**) levels. To identify differentially abundant species from the phylum to genus levels, two methods were employed: Student’s *t*-test (**E**) and the LEfSe bar chart (**F**). Differences were defined as significance with * *p* < 0.05.

**Figure 3 microorganisms-13-02375-f003:**
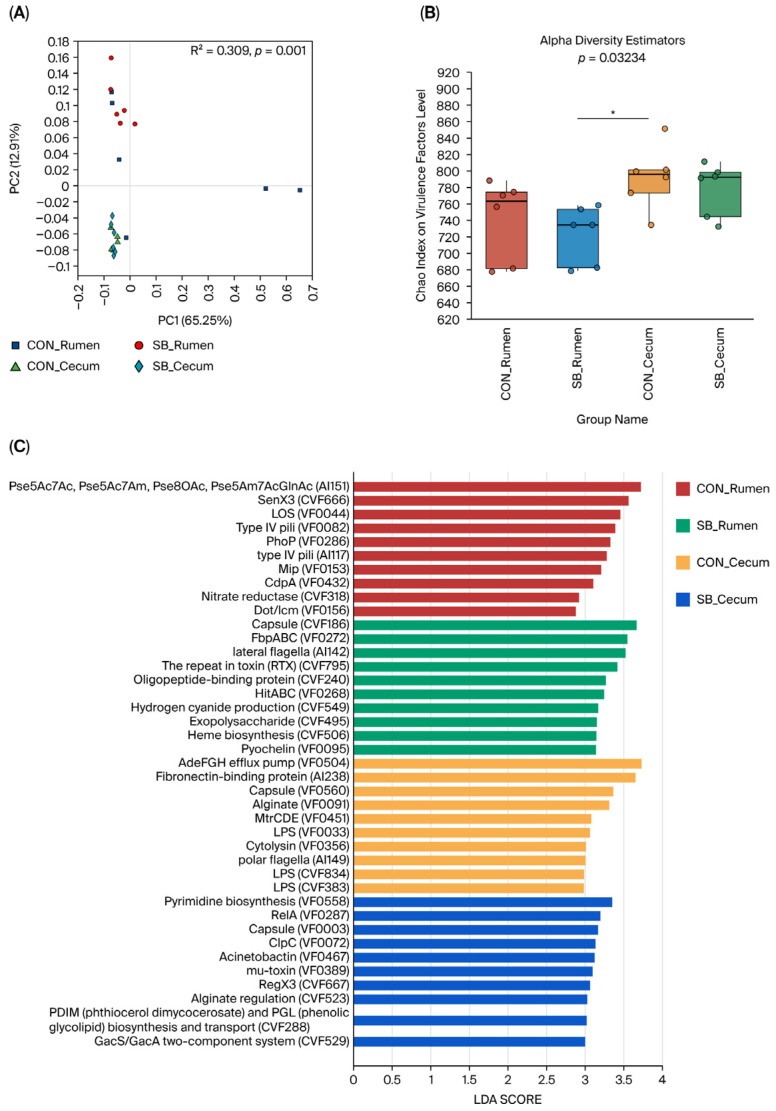
The effect of adding sodium butyrate to milk on rumen and cecum VFGs. (**A**) PCoA based on the Bray–Curtis distance of VFGs in the rumen and cecum of calves. (**B**) The α-diversity of VFGs in the rumen and cecum of cows. (**C**) Identification of signature VFGs in the four groups. Differences were defined as significance with * *p* < 0.05.

**Figure 4 microorganisms-13-02375-f004:**
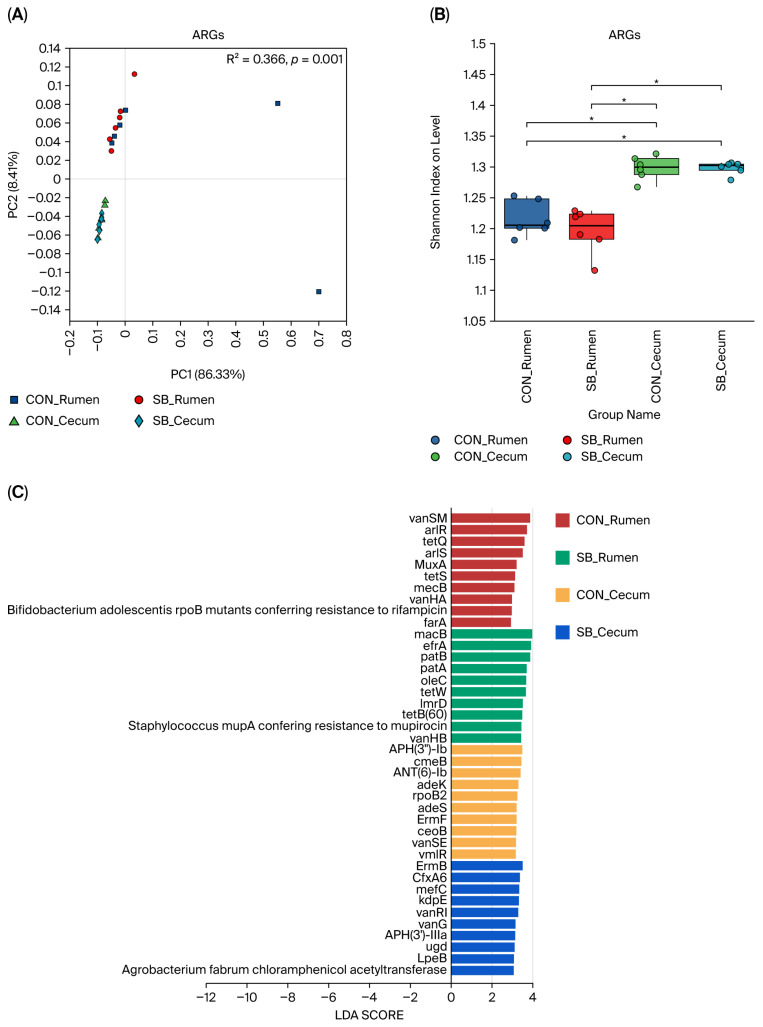
The effect of sodium butyrate addition to milk on rumen and cecum ARGs. (**A**) PCoA based on the Bray–Curtis distance of ARGs in the rumen and cecum of calves. (**B**) The α-diversity of ARGs in the rumen and cecum of cows. (**C**) Identification of signature ARGs in the four groups. Differences were defined as significance with * *p* < 0.05.

## Data Availability

All raw sequencing data that support the findings of this study have been deposited into the Sequence Read Archive (SRA) under accession number PRJNA1274645.

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
