# Peer review of "Sodium Butyrate Supplementation in Whole Milk Modulates the Gastrointestinal Microbiota Without Altering the Resistome and Virulome in Preweaned Calves"

_microorganisms, 2025, doi:10.3390/microorganisms13102375_

Round 1
Reviewer 1 Report
Comments and Suggestions for Authors
Dear Authors,
your study explored the role and the effects of Sodium butyrate feed supplementation in altering the gut microbiota and (ARG) resistome in calves, by means of 16S sequencing and metagenomic analysis. The paper is well structured with a clear description of methodology also for reproducibility.
The topic is relevant in animal science and microbiology such as for a One Health perspective that has to be emphasized along the text, considering the global relevance of ARG nowadays. In addition even if the work is potentially publishable there are some concerns and gaps to address and to fill before its acceptance.
In the Abstract , also the correct metagenomic concept in addition to macrogenomic term is correct to use. All the acronyms have to be defined along the manuscript.
In the Introduction please add data about the study of SB effect on microbiota perturbation in other species or about the implication of other supplementations in calves microbiota regulation.
In Materials and Methods section some information about analysis of virulence factors, bacterial diversity and antibiotic resistance are lacking, as well as about which genes are most relevant for the improvement of calf health, focusing on the main novelty of the study.
Please, add information about animal housing and feeding, important parameters and factors that could condition the outcomes trend.
The major concerns are about the sample size and the statistical power that could be reduced for the use of only 16 calves (6 per group). Please discuss and justify in depth this study limitation.
Many results are statistically significant but the magnitude of the difference, or the effect size, is less clear. For example the increase of Bifidobacterium abundance or decreases in Bacteroides are reported, but the relative abundance (%) should be discussed in terms of biological relevance.
The paper assumes that SB had no effects and implications on VGF and ARG composition, but it is required an evaluation about the dose effect correlation, the administration timeline and the presence of other bacterial disease that could alterate the microbiota and intestinal flora.
In discussion some information are redundant as the comparison with previous SB supplementation studies, that are lacking in Introduction section.
In conclusion revise some figures caption, some too long and other less clear, together with the images resolution. In the results section you can consider to move some list of enriched genes (i.e detailed VGF or ARGs names) to supplementary materials for a better readability.
Comments on the Quality of English LanguageOverall the English language is comprehensive, but some passages in the text could be made more fluent correcting somer grammars imprecisions.
Author Response
Reviewer 1:
your study explored the role and the effects of Sodium butyrate feed supplementation in altering the gut microbiota and (ARG) resistome in calves, by means of 16S sequencing and metagenomic analysis. The paper is well structured with a clear description of methodology also for reproducibility.
The topic is relevant in animal science and microbiology such as for a One Health perspective that has to be emphasized along the text, considering the global relevance of ARG nowadays. In addition even if the work is potentially publishable there are some concerns and gaps to address and to fill before its acceptance.
In the Abstract , also the correct metagenomic concept in addition to macrogenomic term is correct to use. All the acronyms have to be defined along the manuscript.
Response: Thank you for your suggestion. This was due to an error in our writing. We have now standardized all abbreviations throughout the text to make it easier for readers to follow.
In the Introduction please add data about the study of sodium butyrate effect on microbiota perturbation in other species or about the implication of other supplementations in calves microbiota regulation.
Response: Thank you for your suggestions, which is valuable for improving the accuracy of the manuscript. We have added research progress on sodium butyrate in other animals (Line 66-67).
In Materials and Methods section some information about analysis of virulence factors, bacterial diversity and antibiotic resistance are lacking, as well as about which genes are most relevant for the improvement of calf health, focusing on the main novelty of the study.
Response: Thank you for your suggestion. We have made changes (155-166)
.
Please, add information about animal housing and feeding, important parameters and factors that could condition the outcomes trend.
Response: Thank you for your suggestion. We have made changes (see Line 94-110).
The major concerns are about the sample size and the statistical power that could be reduced for the use of only 16 calves (6 per group). Please discuss and justify in depth this study limitation.
Response: Based on Górka et al. [1], the PROC POWER and the GLMPOWER procedure in SAS (SAS Institute Inc., Cary, NC) were used to yield a sample size with a power of 0.8 under P < 0.05. According to previous similar studies, the effect size f that can make a difference in BW, diarrhea frequency, and crypt depth were 0.68 [2], 1.30 [3], and 1.24 [4], respectively. Based on the results of the power analysis, a total of at least 20 calves (10 calves per treatment) were needed to study growth performance and 8 calves (4 calves per treatment) to study health and intestinal development to meet the power requirements when the treatment groups were 2. The 12 male calves that were slaughtered were used to analyze the data of gastrointestinal tract microbiota on d 74. Therefore, selecting 6 replicates per group in this study is reasonable.
- Górka, P.; Kowalski, Z.M.; Pietrzak, P.; Kotunia, A.; Jagusiak, W.; Holst, J.J.; Guilloteau, P.; Zabielski, R. Effect of method of delivery of sodium butyrate on rumen development in newborn calves. Dairy. Sci.2011, 94, 5578-5588.
- Brunton, L.A.; Reeves, H.E.; Snow, L.C.; Jones, J.R. A longitudinal field trial assesing the impact of feeding waste milk containing antibiotic residues on the prevalence of ESBL-producing Escherichia coli in calves. Vet. Med.2014, 117, 403-412.
- Ma, L.; Zhu, Y.; Zhu La, A.L.T.; Lourenco, J.M.; Callaway, T.R.; Bu, D. Schizochytrium sp. And lactoferrin supplementation alleviates Escherichia coli K99-induced diarrhea in preweaning dairy calves. Dairy. Sci.2024, 107, 1603-1619.
- Pyo, J.; Hare, K.; Pletts, S.; Inabu, Y.; Haines, D.; Sugino, T.; Guan, L.L.; Steele, M. Feeding colostrum or a 1:1 colostrum:milk mixture for 3 days postnatal increases small intestinal development and minimally influences plasma glucagon-like peptide-2 and serum insulin-like growth factor-1 concentrations in Holstein bull calves. Dairy. Sci.2020, 103, 4236-4251.
Many results are statistically significant but the magnitude of the difference, or the effect size, is less clear. For example the increase of Bifidobacterium abundance or decreases in Bacteroides are reported, but the relative abundance (%) should be discussed in terms of biological relevance.
Response: We appreciate the reviewer's feedback regarding the limitations of our study in discussing biological relevance, which is indeed an area requiring further refinement. We have made the revisions; please refer to the Discussion section of the manuscript.
The paper assumes that sodium butyrate had no effects and implications on VGF and ARG composition, but it is required an evaluation about the dose effect correlation, the administration timeline and the presence of other bacterial disease that could alterate the microbiota and intestinal flora.
Response: Thank you for your comments. In our previous studies, we evaluated the effects of sodium butyrate dosage and timing, demonstrating that the optimal supplementation level decreases with calf age, reaching 8.78 g/d. This reduction occurs because the gastrointestinal microbiota gradually matures with age, enabling calves to produce butyrate independently. Building upon these prior findings, we evaluated the effects of supplementing sodium butyrate in whole milk on the abundance of resistance genes and virulence factors in the rumen and cecum. However, this study did not find that sodium butyrate supplementation reduced the abundance of resistance genes and virulence factors in the gastrointestinal tract. This may be related to factors such as age and overall health. Future research should further evaluate the effects of sodium butyrate on calves of different ages, administered via different routes, and in varying health conditions.
In discussion some information are redundant as the comparison with previous sodium butyrate supplementation studies, that are lacking in Introduction section.
Response: In the introduction section of the manuscript, we made several content revisions, emphasizing the hazards of resistance genes and virulence factors, as well as current research progress on sodium butyrate. In the discussion section, we integrated our findings with previous studies. Please refer to the introduction and discussion sections of the manuscript.
In conclusion revise some figures caption, some too long and other less clear, together with the images resolution. In the results section you can consider to move some list of enriched genes (i.e detailed VGF or ARGs names) to supplementary materials for a better readability.
Response: Thank you for your suggestions. We have made changes. Please refer to the main text of the manuscript.
Reviewer 2 Report
Comments and Suggestions for Authors
Respectfully, I recognize that the manuscript is an important and relevant topic. However, several critical issues currently impact the overall quality, clarity, reproducibility, and readability of the work - particularly concerning the experimental design. The design of the study is limited, most notably due to the small sample size, with only six animals per treatment group. This number is insufficient to support statistically robust or generalizable conclusions. Additionally, the manuscript suffers from a generic and overly broad writing style, which detracts from the scientific rigor and perceived significance of the findings. A more precise and technically focused approach to both the writing and the experimental justification is necessary to enhance the manuscript’s impact.
Abstract:
Lines 25-27: This is not accurate; while VFG and ARG may have been studied previously, this may be the first study to investigate sodium butyrate effects on VFG and ARG. However, my suggestion is to remove statements such as “Our study is the first,” “Few studies exist,” or “Data are minimal.” Rather than strengthening the manuscript, such expressions can inadvertently signal a lack of scientific novelty. They often appear as an appeal for validation rather than a demonstration of the study’s intrinsic value. This manuscript has several strong and original aspects that can be emphasized more effectively. Therefore, I recommend either removing these statements or rewriting them to focus on the study’s specific contributions. The same applies to similar phrasing in other sections.
Keyword: My suggestion is to use keywords other than the title and review the keyword suggestion in the author's instructions.
Introduction: The introduction lacks clear organization in many sections; thus, it is essential to revise it to ensure a coherent and logical flow of ideas. Enhancing the writing style can be achieved by incorporating numerical data, which provides greater specificity and supports clearer interpretation of the findings.
Lines 39-41: Including a description of the study’s results in the introduction is not recommended, as this section should focus on background information and the study’s objectives.
Lines 47-48: The connection between these paragraphs is unclear. Consider revising them to improve the flow and ensure a logical progression of ideas. Similar suggestion for other parts of the introduction.
Line 81: Add the objective of the study.
Results. The results description is generic. Improve your writing style showing your data in other forms more than a simple description of tables or figures. Was higher? How much (%, g, l, etc)?
Figures 1 and 2: The multivariate analysis presented in Figures 1 and 2 is not described in the Statistical Analysis section of the Materials and Methods. Please ensure that all analyses included in the results are properly described in the methodology.
Lines 228 and 255: The comparison between the cecum and rumen is not within the scope of this study. Although this information may be relevant, it does not align with the study's primary objective, which is to investigate the effect of butyrate.
Discussion: The current discussion is overly speculative in some sections. It should instead focus on interpreting how the results were obtained, supported by biological, metabolic, physiological, or environmental mechanisms. While the existing discussion provides a good general overview, describes results, and compares findings with those of other authors, it lacks a focused interpretation of the mechanisms behind the observed results. To strengthen the discussion, please integrate relevant theories, hypotheses, or mechanisms that could explain the outcomes.
Conclusion: The conclusion requires revision for clarity and conciseness. It should be rewritten in a more direct and focused manner, clearly summarizing the main findings and their implications.
Lines 381–383: These lines do not contribute meaningful information and should be removed.
Line 384: The fixation process was not evaluated in this study and is not aligned with the stated objectives. Consider removing or revising this section accordingly.
Author Response
Reviewer 2:
Respectfully, I recognize that the manuscript is an important and relevant topic. However, several critical issues currently impact the overall quality, clarity, reproducibility, and readability of the work - particularly concerning the experimental design. The design of the study is limited, most notably due to the small sample size, with only six animals per treatment group. This number is insufficient to support statistically robust or generalizable conclusions. Additionally, the manuscript suffers from a generic and overly broad writing style, which detracts from the scientific rigor and perceived significance of the findings. A more precise and technically focused approach to both the writing and the experimental justification is necessary to enhance the manuscript’s impact.
Abstract:
Lines 25-27: This is not accurate; while VFG and ARG may have been studied previously, this may be the first study to investigate sodium butyrate effects on VFG and ARG. However, my suggestion is to remove statements such as “Our study is the first,” “Few studies exist,” or “Data are minimal.” Rather than strengthening the manuscript, such expressions can inadvertently signal a lack of scientific novelty. They often appear as an appeal for validation rather than a demonstration of the study’s intrinsic value. This manuscript has several strong and original aspects that can be emphasized more effectively. Therefore, I recommend either removing these statements or rewriting them to focus on the study’s specific contributions. The same applies to similar phrasing in other sections.
Response: Thank you for your suggestions. This statement did indeed contain inaccuracies. We have removed it and conducted a comprehensive search and revision of the entire text.
Keyword: My suggestion is to use keywords other than the title and review the keyword suggestion in the author's instructions.
Response: Thank you for your suggestion. We have made changes. Please refer to the Keywords section of the manuscript.
Introduction: The introduction lacks clear organization in many sections; thus, it is essential to revise it to ensure a coherent and logical flow of ideas. Enhancing the writing style can be achieved by incorporating numerical data, which provides greater specificity and supports clearer interpretation of the findings.
Response: Thank you for your positive comments and valuable suggestions to improve the quality of our manuscript. We have revised the writing style and content of the introduction section. Please refer to the introduction chapter of the manuscript.
Lines 39-41: Including a description of the study’s results in the introduction is not recommended, as this section should focus on background information and the study’s objectives.
Response: Thank you for your suggestion. We have made changes.
Lines 47-48: The connection between these paragraphs is unclear. Consider revising them to improve the flow and ensure a logical progression of ideas. Similar suggestion for other parts of the introduction.
Response: Thank you for your positive comments and valuable suggestions to improve the quality of our manuscript. We have made changes. Please refer to the introduction chapter of the manuscript.
Line 81: Add the objective of the study.
Response: Thank you for your suggestion. We have made changes. Given this hypothesis, the objective of this study is to directly investigate whether such long - term supplementation can indeed effectively stimulate the GIT microbiota of the experimental subjects and, as a consequence, optimize the distribution of GIT VFGs and ARGs, ultimately providing valuable insights into a potential strategy for regulating microbial - gene dynamics in the GIT.
Results. The results description is generic. Improve your writing style showing your data in other forms more than a simple description of tables or figures. Was higher? How much (%, g, l, etc)?
Response: Thank you for your suggestion. We have revised the content of the Results section. Please refer to the Results section of the manuscript.
Figures 1 and 2: The multivariate analysis presented in Figures 1 and 2 is not described in the Statistical Analysis section of the Materials and Methods. Please ensure that all analyses included in the results are properly described in the methodology.
Response: Thank you for bringing this issue to our attention. This was caused by an error in our writing, and we have already revised the data statistics. Please refer to the manuscript statistics (Line .175-194).
Lines 228 and 255: The comparison between the cecum and rumen is not within the scope of this study. Although this information may be relevant, it does not align with the study's primary objective, which is to investigate the effect of butyrate.
Response: Thank you for your suggestion. We have revised the content of the Results section. Please refer to the Results section of the manuscript.
Discussion: The current discussion is overly speculative in some sections. It should instead focus on interpreting how the results were obtained, supported by biological, metabolic, physiological, or environmental mechanisms. While the existing discussion provides a good general overview, describes results, and compares findings with those of other authors, it lacks a focused interpretation of the mechanisms behind the observed results. To strengthen the discussion, please integrate relevant theories, hypotheses, or mechanisms that could explain the outcomes.
Response: In this study, supplementation with sodium butyrate did not affect the diversity or composition of rumen and cecal ARGs and VFGs, likely because the selected calves were in good health. Additionally, supplementation with sodium butyrate in whole milk directly acts on the gut, which may also contribute to its effects on rumen ARGs and VFGs. Notably, sodium butyrate undergoes rapid degradation and absorption in the gastrointestinal tract, where it is oxidized into ketone bodies. Furthermore, the optimal sodium butyrate supplementation rate decreases with calf age, potentially explaining why supplementation may not yield effects during the pre-weaning period.
Conclusion: The conclusion requires revision for clarity and conciseness. It should be rewritten in a more direct and focused manner, clearly summarizing the main findings and their implications.
Response: Thank you for your suggestion. We have made changes (see line 465-470).
Lines 381–383: These lines do not contribute meaningful information and should be removed.
Response: Thank you for your suggestion. We have deleted it.
Line 384: The fixation process was not evaluated in this study and is not aligned with the stated objectives. Consider removing or revising this section accordingly.
Response: Thank you for your comment. We have deleted it.
Round 2
Reviewer 1 Report
Comments and Suggestions for Authors
Dear Authors,
I appreciate your revision work, beacuse you have made the right corrections and amendements, answering to my observations. Furthermore, you have filled gaps and you have implemented the lacking relevant concept and information. The manuscript was improved in the quality presentation and in the overall comprehension. My advice is to continue to investigate the role of sodium butyrate, also in other species and by other route of administration, to fully comprehend its putative and useful application.
Best regards.
Comments on the Quality of English LanguageThe English could be improved to more clearly express the research.
Author Response
Reviewer 1:
I appreciate your revision work, beacuse you have made the right corrections and amendements, answering to my observations. Furthermore, you have filled gaps and you have implemented the lacking relevant concept and information. The manuscript was improved in the quality presentation and in the overall comprehension. My advice is to continue to investigate the role of sodium butyrate, also in other species and by other route of administration, to fully comprehend its putative and useful application.
Response: Thank you for your suggestion. Thank you for your valuable suggestions. We will incorporate your insights into our subsequent trials. While the effects of sodium butyrate delivery methods on gastrointestinal development in pre-weaned calves have been documented, there is currently no research on how these delivery methods influence microbial communities, virulence factors, or resistance genes. Therefore, your suggestion is highly valuable. We will proceed with further trials examining sodium butyrate delivery methods and their application across different animal species.
In this study, we investigated the effects of SB supplementation on rumen and cecum microbiota, VFGs, and ARGs in preweaned calves. We found no significant impact of SB supplementation on virulence factors or resistance genes, but it did promote the colonization of beneficial microorganisms in the rumen and cecum. These findings may be limited by the quality of colostrum received by calves and the small sample size. Future research plans will further evaluate nutritional strategies (such as supplementing with Bifidobacterium) to reduce the abundance of gastrointestinal pathogens and resistance genes in pre-weaning calves, providing theoretical support for healthy dairy cattle management. Additionally, this study found that supplementing milk with sodium butyrate affects rumen development, necessitating future investigation into the mechanisms underlying rumen-gut interactions.
Reviewer 2 Report
Comments and Suggestions for Authors
Dear Authors,
Thank you for the revisions made to the manuscript. In my view, the writing style has improved. However, I am unable to recommend the manuscript for publication due to statistical concerns, particularly the low number of experimental units, which compromises the reliability and reproducibility of the results.
Author Response
Reviewer 2:
Thank you for the revisions made to the manuscript. In my view, the writing style has improved. However, I am unable to recommend the manuscript for publication due to statistical concerns, particularly the low number of experimental units, which compromises the reliability and reproducibility of the results.
Response:
We thank the reviewer for raising this important point regarding the sample size in our study. We acknowledge that the statistical power of any experiment is influenced by the number of biological replicates, and we agree that this is a key consideration in interpreting our results.
Based on Górka et al. [1], the PROC POWER and the GLMPOWER procedure in SAS (SAS Institute Inc., Cary, NC) were used to yield a sample size with a power of 0.8 under P < 0.05. According to previous similar studies, the effect size f that can make a difference in BW, diarrhea frequency, and crypt depth were 0.68 [2], 1.30 [3], and 1.24 [4], respectively. Based on the results of the power analysis, a total of at least 20 calves (10 calves per treatment) were needed to study growth performance and 8 calves (4 calves per treatment) to study health and intestinal development to meet the power requirements when the treatment groups were 2. The 12 male calves that were slaughtered were used to analyze the data of gastrointestinal tract microbiota on d 74. Therefore, selecting 6 replicates per group in this study is reasonable.
The choice of a sample size of 6 animals per group was based on several practical and ethical considerations inherent to slaughter trials with neonatal calves. To minimize confounding variables, we utilized highly homogeneous calves in terms of breed, age, birth weight, and management history. Sourcing a large number of such uniform neonatal animals for a terminal study is logistically challenging and extremely costly. This study is one of the first to concurrently profile the microbiome, resistome, and virulome in both the rumen and cecum of preweaned calves. Its primary goal was to identify prominent patterns and generate hypotheses for larger-scale, confirmatory studies in the future.
We have added a paragraph in the Discussion section (Line 458-468) that acknowledges the limited sample size and cautions against over-interpretation of negative results (i.e., trends that did not reach statistical significance). We also state that our findings, while promising, require validation in larger cohorts.
- Górka, P.; Kowalski, Z.M.; Pietrzak, P.; Kotunia, A.; Jagusiak, W.; Holst, J.J.; Guilloteau, P.; Zabielski, R. Effect of method of delivery of sodium butyrate on rumen development in newborn calves. Dairy. Sci.2011, 94, 5578-5588.
- Brunton, L.A.; Reeves, H.E.; Snow, L.C.; Jones, J.R. A longitudinal field trial assesing the impact of feeding waste milk containing antibiotic residues on the prevalence of ESBL-producing Escherichia coli in calves. Vet. Med.2014, 117, 403-412.
- Ma, L.; Zhu, Y.; Zhu La, A.L.T.; Lourenco, J.M.; Callaway, T.R.; Bu, D. Schizochytrium sp. And lactoferrin supplementation alleviates Escherichia coli K99-induced diarrhea in preweaning dairy calves. Dairy. Sci.2024, 107, 1603-1619.
Pyo, J.; Hare, K.; Pletts, S.; Inabu, Y.; Haines, D.; Sugino, T.; Guan, L.L.; Steele, M. Feeding colostrum or a 1:1 colostrum:milk mixture for 3 days postnatal increases small intestinal development and minimally influences plasma glucagon-like peptide-2 and serum insulin-like growth factor-1 concentrations in Holstein bull calves. J. Dairy. Sci. 2020, 103, 4236-4251.